# Extracellular Vesicles-Induced Cell Homing and Odontogenesis via microRNA Signaling for Dentin Regeneration

**DOI:** 10.3390/ijms26157182

**Published:** 2025-07-25

**Authors:** Venkateswaran Ganesh, Douglas C. Fredericks, Emily B. Petersen, Henry L. Keen, Rui He, Jordon D. Turner, James A. Martin, Aliasger K. Salem, Kyungsup Shin, Abhishek Parolia, Dongrim Seol

**Affiliations:** 1Department of Orthopedics and Rehabilitation, Carver College of Medicine, University of Iowa, Iowa City, IA 52242, USA; venkateswaran-ganesh@uiowa.edu (V.G.); douglas-fredericks@uiowa.edu (D.C.F.); emily-petersen@uiowa.edu (E.B.P.); jordon-turner@uiowa.edu (J.D.T.); james-martin@uiowa.edu (J.A.M.); 2Department of Roy J. Carver Biomedical Engineering, College of Engineering, University of Iowa, Iowa City, IA 52242, USA; 3Iowa Institute of Human Genetics, University of Iowa, Iowa City, IA 52242, USA; henry-keen@uiowa.edu; 4Pharmaceutical Sciences and Experimental Therapeutics, College of Pharmacy, University of Iowa, Iowa City, IA 52242, USA; rui-he@uiowa.edu (R.H.); aliasger-salem@uiowa.edu (A.K.S.); 5Department of Orthodontics, College of Dentistry and Dental Clinics, University of Iowa, Iowa City, IA 52242, USA; kyungsup-shin@uiowa.edu; 6Department of Endodontics, College of Dentistry and Dental Clinics, University of Iowa, Iowa City, IA 52242, USA; abhishek-parolia@uiowa.edu

**Keywords:** extracellular vesicles, odontogenesis, dentin regeneration, dentinogenesis, microRNA, cell homing, dental pulp stem cells

## Abstract

Reparative tertiary dentinogenesis requires the recruitment and odontogenic differentiation of dental pulp stem cells (DPSCs). Extracellular vesicles (EVs) as bioactive molecules have gained attention in regenerative medicine for their ability to mediate tissue repair through intercellular communication, influencing cell recruitment, proliferation, and differentiation. This study aimed to evaluate the effects of EVs on DPSC homing and odontogenic differentiation for dentin regeneration. DPSC-derived EVs were cultured in either growth (EV-G) or odontogenic differentiation (EV-O) conditions and isolated using a modified precipitation method. EVs were characterized by nanoparticle tracking analysis, scanning electron microscopy, antibody array, and cellular uptake assay. Treatment with 5 × 10^8^ EVs/mL significantly enhanced DPSC chemotaxis and proliferation compared with a no-treatment control and a lower dosage of EV (5 × 10^7^ EVs/mL). Gene expression and biochemical analyses revealed that EV-O up-regulated odontogenic markers including collagen type 1A1 (COL1A1), runt-related transcription factor 2 (RUNX2), and alkaline phosphatase (ALP). EV-O enhanced dentin regeneration by approximately 55% over vehicle controls in a rabbit partial dentinotomy/pulpotomy model. We identified key microRNAs (miR-21-5p, miR-221-3p, and miR-708-3p) in EV-O involved in cell homing and odontogenesis. In conclusion, our EV-based cell homing and odontogenic differentiation strategy has significant therapeutic potential for dentin regeneration.

## 1. Introduction

Dentin is a porous, mineralized tissue beneath the enamel that primarily protects the pulp-dentin complex. Odontoblasts, located in the outer layer of the dental pulp, are responsible for dentin formation and maintenance. When damaged, they are replaced by odontoblast-like cells derived from dental pulp stem cells (DPSCs) [1,2]. Therefore, DPSC homing and odontogenic differentiation via bioactive molecules are essential for reparative tertiary dentinogenesis [3,4].

Extracellular vesicles (EVs) including exosomes (30–150 nm size range), microvesicles (MVs; 100–1000 nm size range), and apoptotic bodies have attracted attention in regenerative medicine for their remarkable potential to promote cell recruitment, proliferation, and differentiation through intercellular communication [5,6,7,8,9]. In the previous study, EVs isolated from odontogenically differentiated human DPSCs (EV-O) elevated the expression of odontogenic genes in vitro through transforming growth factor beta 1 (TGFβ1)/mothers against decapentaplegic homolog (Smad) pathway [10]. However, the effects of EV-O on cell homing were not explored [10]. Our previous studies revealed that EVs derived from DPSCs under growth (EV-G) and angiogenic differentiation conditions (EV-A) promoted approximately a 4-fold increase over control in DPSC migration [11]. Thus, EV-O can be a great source to enhance both DPSC homing and odontogenesis for pulpodentin regeneration.

Among a variety of bioactive molecules in EVs such as deoxyribonucleic acid (DNA), ribonucleic acid (RNA), lipids, and proteins [12], microRNAs (miRNAs/miRs) play a crucial role in regulating gene expression and cellular processes [13]. During tooth development or repair, several miRNAs are strongly associated with odontogenic differentiation for dentin formation [10,14,15]. For example, miR-27a-5p mimics over-expressed in human EV-O up-regulated odontogenic markers, dentin sialophosphoprotein (DSPP), dentin matrix protein 1 (DMP-1), alkaline phosphatase (ALP), and runt-related transcription factor 2 (RUNX2) [10]. Thus, miRNA sequencing can allow understanding of the functional mechanisms of EVs and targeting potential miRNAs.

We hypothesized that EV-O has unique miRNA profiles to enhance DPSC chemotaxis and odontogenesis. The objectives of this study were to evaluate the effects of DPSC-EVs and to compare EV-G and EV-O on cell homing and odontogenesis for dentin regeneration via both in vitro and in vivo studies.

## 2. Results

### 2.1. Characterization of DPSC-EVs

The characterization of EVs was performed by nanoparticle tracking analyzer (NTA), scanning electron microscopy (SEM), and antibody array. The concentration of EV-G (1.6 × 10^11^ particles/mL) was approximately 3.8-fold higher than that of EV-O (3.06 × 10^10^ particles/mL), while the mean size of EV-O was approximately 2-fold larger than that of EV-G (206.2 nm EV-O versus 104.3 nm EV-G: *p* < 0.001) (Figure 1A,B). The spheroid-like morphology of EVs was visually validated in a SEM image (Figure 1C). EV-O showed highly positive expression of annexin A5 (ANXA5), tumor susceptibility gene 101 (TSG101), flotillin-1 (FLOT1), intercellular adhesion molecule 1 (ICAM), programmed cell death 6 interacting protein (ALIX), and CD81 as EV surface markers, and no detectable expression of cis-Golgi matrix protein (GM130) as a cellular contamination marker during EV isolation (Figure 1D).

### 2.2. In Vitro Efficacies of EV-O on Cell Viability, Proliferation, Chemotaxis, and Odontogenesis

We confirmed that both EV-G and EV-O labeled with PKH67 green fluorescence were highly internalized into the DPSCs in an uptake assay (Figure 2A). Next, a viability test was conducted to assess any cytotoxic effect of DPSC-EV in various concentrations. There was no sign of cell death in either the vehicle or the no EV controls; therefore, all data were pooled together (labeled No EV) (Figure 2B). Out of three EV concentrations, 5 × 10^9^ particles/mL EV-G (EV-G-H) showed a significant decrease in cell viability (*p* = 0.004 versus No EV) (Figure 2B). Due to cytotoxicity observed at the highest concentrations, EV-G-H and EV-O-H were excluded for further evaluation. After 4 days of treatment, the medium concentration (5 × 10^8^ particles/mL) of EVs induced a slight increase of cell proliferation in both EV-G-M (*p* < 0.001 versus No EV or EV-G-L) and EV-O-M (*p* < 0.001 versus No EV or EV-O-L) (Figure 2C). In a chemotactic assay, DPSC-EV treatment with a concentration of 5 × 10^8^ particles/mL promoted a significant increase of DPSC migration in both groups of EV-G (*p* < 0.001 versus No EV or EV-G-L) and EV-O (*p* < 0.001 versus No EV or EV-O-L) (Figure 2D,E). In contrast, there was no effect of low concentration of EVs (5 × 10^7^ particles/mL).

DPSCs were treated with 5 × 10^8^ particles/mL EVs in basal or complete odontogenic induction media (bOM or cOM, respectively) to evaluate the effect of DPSC-EVs on odontogenic differentiation for 10 days. Collagen type 1A1 (COL1A1) was significantly up-regulated by the presence of EV-O in both bOM (*p* = 0.025 versus No EV and *p* = 0.005 versus EV-G) and cOM (*p* < 0.001 No EV pr EV-G) (Figure 2F). Similarly, EV-O induced higher expression of RUNX2 in only cOM (*p* < 0.001 versus No EV and *p* = 0.006 versus EV-G) (Figure 2G). Thus, the odontogenic effects of EV-O were additive in the cOM condition. Moreover, ALP content was slightly higher when treated with EV-G (*p* = 0.014 versus No EV) or EV-O (*p* = 0.002 versus No EV) (Figure 2H).

### 2.3. miRNA Profiles

In NGS analysis, we identified candidate miRNAs that regulate cell homing and odontogenesis in DPSC-EVs. A total of 474 mature and 254 hairpin miRNAs in EV-G and 431 mature and 256 hairpin miRNAs in EV-O were detected. Table 1 shows a list of the 30 most prevalent mature miRNAs. In particular, 4 miRNAs (ocu-miR-146a-5p: 13.32%, ocu-miR-199a-3p: 9.80%, ocu-miR-122-5p: 8.57%, ocu-miR-221-3p: 8.32%) and 3 miRNAs (ocu-miR-21-5p: 17.51%, ocu-miR-199a-3p: 10.77%, ocu-miR-221-3p: 10.28%) represented more than 8% of the total reads in EV-G and EV-O, respectively. All differentially expressed mature miRNAs are listed in Table 2 and plotted in Appendix A. EV-O isolated under odontogenic induction showed 35 up-expressed (ocu-miR-708-5p/3p, ocu-miR-885-5p, ocu-miR-24-2-5p, ocu-miR-29c-3p, etc.) and 38 down-expressed (ocu-miR-146a-5p, ocu-miR-122-5p, ocu-miR-503-5p, ocu-miR-16b-5p, ocu-miR-378-3p, etc.) miRNAs among over 0.01% population. Thirty-six hairpin miRNAs significantly changed in EV-O, of which 21 hairpin miRNAs (ocu-miR-708, ocu-miR-509c, ocu-miR-498, etc.) increased and 15 hairpin miRNAs (ocu-miR-146a, ocu-miR-503, ocu-miR-122, etc.) decreased (Appendix A). The top 3 miRNAs with significant expression of both mature and hairpin miRNAs in EV-O were ocu-miR-708 (up), ocu-miR-328 (up), ocu-miR-29b (up), ocu-miR-146a (down), ocu-miR-503 (down), and ocu-miR-122 (down) (Appendix A).

### 2.4. In Vivo Efficacies

At 4 weeks, both injured maxillary and intact mandibular incisors were harvested for histological examination. Initially, the degree of tissue damage through partial dentinotomy/pulpotomy was validated in the images of hematoxylin and eosin staining. Compared with intact control in the deep area around the pulp (Figure 3E,F), vehicle control showed no apparent difference in dentin bridge formation, odontoblast layer, inflammatory cell response, and blood vessel formation (Figure 3H,I). In contrast, the dentin repair was not completed in the incisal area of vehicle control (Figure 3G,J). Inflammatory responses were observed through the pulp cavity trace (PCT) (Figure 3J). In representative images, dentin integrity was improved in EV treatment groups (Figure 3K for EV-G and Figure 3L for EV-O), especially in EV-O. The degree of dentin regeneration in the incisal area was further quantified according to a modified histological scoring system. The reliability of scoring was validated by inter-observer and intra-observer correlation coefficients which were in the excellent range (> 0.7) (Figure 3M). In the scoring comparison, EV-O treatment induced approximately 55% improved dentin regeneration compared to vehicle control (*p* = 0.028) (Figure 3N).

## 3. Discussion

Our findings provide evidence that DPSC-EVs, particularly EV-O, have the potential to activate key processes in dentin repair by promoting cell homing and differentiation. DPSC-EVs (both EV-G and EV-O) significantly promoted DPSC proliferation (Figure 2C), chemotaxis (Figure 2D,E), and odontogenic differentiation (Figure 2F–H) at 5 × 10^8^ particles/mL. In a rabbit partial dentinotomy model, EV-O treatment showed the potential of repairing dentin damage in the incisors (Figure 3).

DPSC-EVs isolated by a polymer-based precipitation method [16] were characterized for size distribution and concentration (Figure 1A,B), morphology (Figure 1C), EV-specific protein markers (Figure 1D), and cellular uptake (Figure 2A) according to a standard guideline from the International Society of Extracellular Vesicles (ISEV) [17]. In NTA, there were apparent differences in size distribution and concentration between EV-G and EV-O (Figure 1A,B). EV-O had a larger mean size (206.2 ± 3.6 nm) than EV-G (104.3 ± 0.6 nm), while EV-G had a higher concentration (1.16 × 10^11^ particles/mL in EV-G versus 3.06 × 10^10^ particles/mL in EV-O). These results suggest that the main population of EV-G and EV-O could be exosomes and MVs, respectively. Although they share functional similarities for tissue regeneration, their differences in origin, size, and cargo may lead to distinct effects on tissue repair processes [18]. For example, our previous work revealed that platelet-rich plasma-derived EVs isolated from early osteoarthritis patients differed in concentration and size and that the larger average EV size correlated with treatment efficacy [19]. Further research is needed to fully understand their mechanisms and to optimize their potential in regenerative medicine. Other EV characteristics, including spheroid-like morphology, EV-specific markers, and cellular uptake, were similar between groups.

Cell homing is an innovative and attractive strategy for dentin regeneration. This strategy aims to recruit endogenous DPSCs to the injured site via chemotactic signals such as basic fibroblast growth factor (bFGF) [20,21], vascular endothelial growth factor (VEGF) [22,23], and granulocyte-colony stimulating factor (G-CSF) [24], and offers an alternative to cell transplantation, which requires cell isolation and manipulation with a higher risk of immune rejection [25]. EVs can also play a critical player in cell homing through directional sensing, cell adhesion, extracellular matrix (ECM) degradation, and leader-follower behavior [26]. In our chemotaxis study, both EV-G and EV-O treatment at 5 × 10^8^ particles/mL enhanced DPSC recruitment by more than 3.5-fold compared to the control (Figure 2D,E). In particular, specific miRNAs as one of the cargoes in EVs can also be utilized to promote cell homing. In previous studies, miR-21-5p and miR-152-3p accelerated diabetic wound healing by enhancing fibroblast migration through the Wnt/β-catenin and phosphatase and tensin homolog (PTEN) signaling pathways [27,28,29]. Similarly, the chemotactic activity of mesenchymal stem cells (MSCs) was promoted by miR-221-3p [30], miR-214-3p [31], and miR-29-3p [32] for tissue repair. Consistent with previous findings, ocu-miR-21-5p (5.52% in EV-G and 17.51% in EV-O), ocu-miR-221-3p (8.32% in EV-G and 10.28% in EV-O), ocu-miR-214-3p (1.92% in EV-G and 4.71% in EV-O), ocu-miR-29a-3p (1.07% in EV-G and 3.58% in EV-O), and ocu-miR-152-3p (1.29% in EV-G and 1.07% in EV-O) were highly expressed in both EV-G and EV-O (Table 1).

Abundant evidence has suggested that EVs derived from odontogenic stem cells induce DPSC differentiation into odontoblasts with the up-regulation of ALP, RUNX2, COL1A1, and bone morphogenetic protein 9 (BMP9) [10,33]. Our results showed similar trends: approximately 3-fold increase in COLA1 (Figure 2F), 2-fold increase in RUNX2 (Figure 2G), and 55% improvement in in vivo dentin formation (Figure 3N). However, our miRNA profiles differed from Hu and colleagues who used human DPSCs as an EV source [10]. The expression of miR-27a-5p, which promoted odontogenic differentiation of DPSCs through transforming growth factor beta 1 (TGFβ1), was less than 0.01% reads in our dataset, compared to 11-fold higher in their study [10]. Instead, our novel findings were significant up-regulation of ocu-miR-708-3p (4.51-fold higher log2 fold change in EV-O: 4.51Δ) [34,35], ocu-miR-22-3p (1.42Δ) [36], ocu-miR-21-5p (1.38Δ) [37], and ocu-miR-29b-3p (1.11Δ) [38,39,40] in EV-O (Table 2), and these miRNAs are known to promote odontogenesis/osteogenesis and to exert anti-inflammation in the previous studies. Among the candidates, ocu-miR-708 and ocu-miR-29b are especially promising due to their differential expressions in both mature and hairpin miRNA profiles (Appendix A).

A rabbit partial dentinotomy/pulpotomy model was used to assess the in vivo efficacy of DPSC-EVs in the incisors [41]. At 4 weeks, intrinsic repair of the pulp-dentin complex was observed, although there were slight differences between intact and vehicle controls regarding the thickness of odontoblast layers and the distribution of blood vessels (Figure 3E–I). One of the possible reasons may be the continuous growth of rabbit incisors (approximately 2 mm per week) with open roots, where numerous stem cells are present [42]. This factor may lead to an overestimation of regeneration potential compared to human teeth. Therefore, future studies should assess earlier time-points (1 or 2 weeks). Unlike the apical area, we could observe partial repair with inflammatory responses at the incisal area (Figure 3J–L). In the histologic grading, the degree of dentin regeneration at the incisal area was significantly improved in the DSPC-EV-treated groups, especially in EV-O (Figure 3N).

In summary, this study demonstrated the odontogenic potential of DPSC-EVs through both in vitro and in vivo approaches and performed miRNA profiling to identify candidates for future therapeutic use. Limitations of our research include no EV characterization of particle/protein ratio and yield per cell, a small sample size of NGS, a lack of NGS data validation using Sanger sequencing or quantitative reverse transcription polymerase chain reaction (qRT-PCR) approaches [43], insufficient replicates for power analysis in the animal studies, and the absence of early time-point (s) in the animal study in the pulp-dentin complex. Future work will explore signal transduction pathways and the efficacies of candidate miRNAs in dentin regeneration.

## 4. Materials and Methods

### 4.1. Isolation of DPSCs

Dental pulp tissues obtained from 2 New Zealand White rabbit cadavers (~10 months old) were minced into approximately 1 mm^3^ pieces and cultured in alpha minimum essential medium (α-MEM; Thermo Fisher Scientific, Waltham, MA, USA) supplemented with 10% fetal bovine serum (FBS; Thermo Fisher Scientific), 50 µg/mL L-ascorbic acid 2-phosphate trisodium salt (FUJIFILM Wako Chemicals, Richmond, VA, USA), 100 U/mL penicillin-streptomycin (Thermo Fisher Scientific), and 2.5 µg/mL amphotericin B (Sigma-Aldrich, St. Louis, MO, USA) in hypoxic culture condition (5% O_2_/CO_2_ at 37 °C). The migrated cells were collected and examined for stem cell characteristics by multipotential differentiation (angiogenesis and odontogenesis) as in our previous studies [11].

### 4.2. Isolation and Characterization of DPSC-EVs

For EV-O isolation, the cells were cultured with odontogenic differentiation medium (ScienCell^TM^ Research Laboratories, Carlsbad, CA, USA) for 20 days. A regular growth medium was used for EV-G isolation. Each conditioned medium (CM) was collected after replacing with α-MEM growth medium containing 10% EV-depleted FBS for 48 h, and DPSC-EVs were isolated by a modified precipitation method using ExoQuick-TC^TM^ (System Biosciences, Palo Alto, CA, USA) [11]. The characterization of DPSC-EVs was examined by NTA using a NanoSight instrument (Malvern Panalytical, Westborough, MA, USA), SEM (S-4800, Hitachi High-Tech, Ibaraki, Japan), antibody array (Exo-Check^TM^, System Biosciences), and cellular uptake using a PKH67 green lipid membrane dye (Sigma-Aldrich) according to previous protocols [11].

### 4.3. Cell Viability, Proliferation, and Chemotaxis

For cell viability, DPSCs seeded at a density of 1 × 10^4^ (200 µL) in 96-well plates were treated with 3 concentrations of DPSC-EVs (EV-G-L and EV-O-L: 5 × 10^7^/mL, EV-G-M and EV-O-M: 5 × 10^8^/mL, or EV-G-H and EV-O-H: 5 × 10^9^/mL) in serum-free culture medium. Because the final products of EVs were diluted in PBS, vehicle controls (0%, 4.3%, 9.0%, and 16.3% (*v*/*v*) PBS) were prepared. The viability was determined by CellTiter 96^®^Aqueous One Solution (Promega, Madison, WI, USA) at 24 h [11]. Similarly, cell proliferation was evaluated using the same method at 4 days. The chemotactic effect of DPSC-EVs was performed using 24-well Transwell^®^ plates with 8 µm-pore polycarbonate membrane inserts (Corning, Corning, NY, USA). Serum-free medium or DPSC-EVs was added to the reservoirs. Following 48 h incubation and washing, the inserts were stained with Calcein AM (1:1000 dilution; Thermo Fisher Scientific), and the migrated cells were digested in papain digestion buffer (1 mg/mL papain, 5 mM L-cysteine hydrochloride acid, 100 mM disodium hydrogen phosphate, 5 mM ethylenediaminetetraacetic acid salt) for Quant-iT^TM^ PicoGreen^®^ dsDNA assay (Thermo Fisher Scientific) at 480 nm excitation and 520 nm emission.

### 4.4. Odontogenesis

The effect of DPSC-EVs (5 × 10^8^/mL) on odontogenic differentiation was evaluated. DPSCs (seeding density: 1.5 × 10^5^/mL) were seeded in 6-well plates and treated with or without 2 mL of the following induction media: (1) basal odontogenic (bOM) differentiation medium without growth factors, (2) bOM + EV-G, (3) bOM + EV-O, (4) complete odontogenic (cOM) differentiation medium with growth factors, (5) cOM + EV-G, and (6) cOM + EV-O. At 10 days, the cells and CM were collected for qRT-PCR [11] and ALP assay, respectively. Species-specific premade probes (Thermo Fisher Scientific) labeled with a fluorescein (FAM) reporter and a minor groove binder (MGB) quencher were used for housekeeping (GAPDH: Oc03823402_g1) and target genes, COL1A1 (Oc03396073_g1) and RUNX2 (Oc02386741_m1). The relative changes in gene expression levels were calculated by the comparative C_T_ (∆∆C_T_) method [44]. For ALP, CMs from cOM were collected and measured 4-methylumbelliferyl phosphate disodium salt (MUP) substrate using a fluorometric ALP assay kit (AB83371; Abcam Limited, Cambridge, UK) at 360 nm excitation and 440 nm emission according to the manufacturer’s instructions.

### 4.5. Next-Generation Sequencing (NGS)

RNAs were extracted from biological duplicates of rabbit DPSC-EVs (EV-G and EV-O) and quantified by a Bioanalyzer Small RNA Assay kit (Agilent, Santa Clara, CA, USA). NGS libraries were prepared and sequenced on a HiSeq^®^ Sequencing System (Illumina, San Diego, CA, USA) with 150 bp paired-end reads at an approximate depth of 10-15 million reads per sample in System Biosciences [11,45]. Raw data were analyzed in the Bioinformatics Division of the Iowa Institute of Human Genetics (IIHG) according to the previous protocol [11]. In brief, the workflow used Trim Galore! (version 0.6.6) for adapter trimming and Bowtie1 (version 1.3.0) for aligning reads to small RNA sequences. Read counts were imported into R (version 3.5.2). Count normalization and statistical analysis were performed using DESeq2 (version 1.22.2).

### 4.6. Rabbit Partial Dentinotomy/Pulpotomy Model

The animal protocol for this study was reviewed and approved by the Institutional Animal Care and Use Committee (IACUC: #1052392) at the University of Iowa. A total of 8 New Zealand White male rabbits (9 months old) were purchased from Envigo (Denver, PA, USA) and acclimated for 10 days before surgery. All animals were housed individually in standard cages under a 12/12 h light/dark cycle in a controlled environment of temperature (18–20 °C), humidity (30–50%), and free access to water and diet. All animals were sedated by an intraperitoneal injection of ketamine HCl (27.5 mg/kg), acepromazine (0.17 mg/kg), xylazine HCl (0.83 mg/kg), and buprenorphine (0.04 mg/kg). Prior to surgery, sustained-release buprenorphine (0.3 mg/kg, subcutaneous (SC)) was administered. A ketamine HCl controlled rate infusion (CRI) (0.04 mg/mL) was administered throughout surgery at a rate of 0.2 mg/kg/hour. Cefazolin (20 mg/kg, intravenous (IV)) was administered pre- and post-operatively as antibiotic prophylaxis. Meloxicam (0.3–0.5 mg/kg, SC) was administered post-operatively and for 10 days following surgery. Lidocaine (2%, 0.5 mL) was infused into the soft tissue surrounding the surgical site following surgical incision closure. General anesthesia was maintained with isoflurane in oxygen via endotracheal intubation at a rate of 1–5%.

A total of 15 maxillary incisors were randomly assigned for vehicle control (injury with HG), EV-G (injury with EV-G/HG), and EV-O (injury with EV-O/HG) (*n* = 5/group). The mandibular incisors were used for an intact control. The head was stabilized on a standard rabbit dental table (Figure 3A), and a class V cavity was created by a dental bur (Figure 3B). Prior to the start of the study, the approach was validated using radiography examination (Figure 3C). The cavities were filled with a mixture of fibrin and hyaluronate hydrogel with or without EV-G or EV-O. The final concentrations of the hydrogel components and EVs are as follows: 12.5 mg/mL fibrinogen and 10 U/mL thrombin (TISSEL^TM^, Baxter, Deerfield, IL, USA), 2.5 mg/mL hyaluronate (GelOne^®^, Zimmer, Warsaw, IN, USA), and 1 × 10^10^ EV particles/mL. After injecting hydrogel, the cavity was capped with resin-modified glass ionomer cement (FujiCEM^TM^ 2, GC International AG, Luzern, Switzerland) (Figure 3D), and surgical incisions were closed with resorbable suture material. Animals were monitored at least twice per day until euthanasia for infection/inflammation, attitude, pain, eating/drinking habits, and wound healing, and were euthanized using isoflurane sedation (1–5% in oxygen) and IV injection of Euthasol (120 mg/kg) at 4 weeks.

### 4.7. Histological Evaluation of Dentinogenesis

The incisors were fixed in 10% (*v*/*v*) buffered neutral formalin solution, decalcified in 5% buffered formic acid, embedded with paraffin, and sagittally sectioned with a 5-micron thickness. The sections were then stained with hematoxylin and eosin (H&E). Two independently blinded observers (V.G. and D.S.) scored twice, with at least one week’s intervals, according to a modified scoring system (Table 3) [41,46]. The scales ranged from 0 to 6 points, and higher points indicated severely damaged dentin. The reliability of the histology grading system was evaluated for inter-observer (between observers) and intra-observer (between two scores from one observer) correlation coefficients using Kendall’s τ-b test (>0.7: excellent, 0.501–0.7: good, 0.301–0.5: moderate, ≤0.3: low) [47].

### 4.8. Statistics

The scatter plots were expressed as the mean values with the standard deviation using GraphPad Prism (Version 10.4.2; San Diego, CA, USA). Parametric data from in vitro studies were compared by one-way ANOVA with the Tukey post-hoc test using SPSS Statistics software (Version 29; IBM, Armonk, NY, USA). Nonparametric data from the histological scoring system for dentin regeneration were analyzed using the Kruskal-Wallis test on ranks with Dunn-Bonferroni post-hoc pairwise comparisons. Statistical significance was set at *p* < 0.05.

## 5. Conclusions

This study aimed to evaluate the effects of DPSC-EVs on cell homing and odontogenic differentiation for dentin regeneration. Our findings demonstrated that EV-O significantly enhanced cell proliferation, chemotaxis, and odontogenic differentiation in both in vitro and in vivo models. In addition, we identified candidate miRNAs that regulate cell homing (miR-21-5p and miR-221-3p) and odontogenesis (miR-708-3p). Thus, our EV-based strategy holds therapeutic potential for dentin regeneration by promoting both cell recruitment and odontogenesis.

## 6. Patents

The authors declare that there is a potential intellectual property concern regarding the publication of this article.

## Figures and Tables

**Figure 1 ijms-26-07182-f001:**
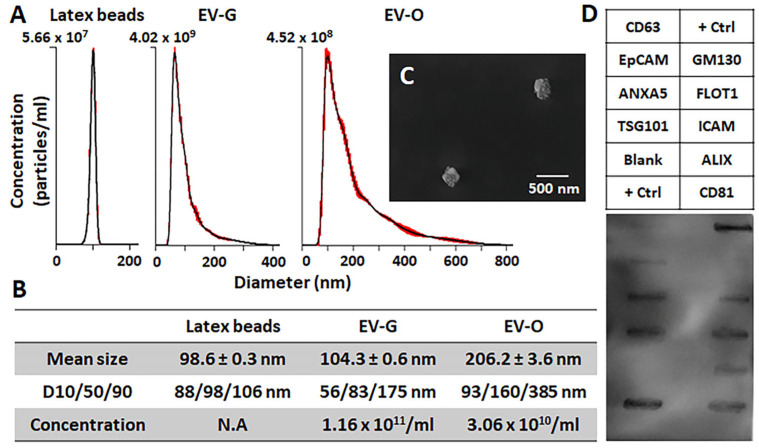
Characterization of rabbit dental pulp stem cell-derived extracellular vesicles (DPSC-EVs). DPSC-EVs were prepared from regular growth medium (EV-G) or odontogenic differentiation medium (EV-O). (**A**) Nanoparticle tracking Analysis (NTA) (*n* = 3). Red error bars indicate ±1 standard error of the mean. Latex beads (100 nm diameter) were used as a reference particle. (**B**) Mean size, distribution, and concentration of EVs. N.A.: not applicable. (**C**) Scanning electron microscopy (SEM) of EV-O. (**D**) Antibody array having eight positive markers (CD63, EpCAM: epithelial cell adhesion molecule, ANXA5: annexin A5, TSG101: tumor susceptibility gene 101, FLOT1: flotillin-1, ICAM: intercellular adhesion molecule 1, ALIX: programmed cell death 6 interacting protein, and CD81) and four controls (cis-Golgi matrix protein (GM130) as a negative marker, 2 positive controls (+Ctrl), and a blank control). The images of SEM and antibody array for EV-G are available in our previous manuscript [11].

**Figure 2 ijms-26-07182-f002:**
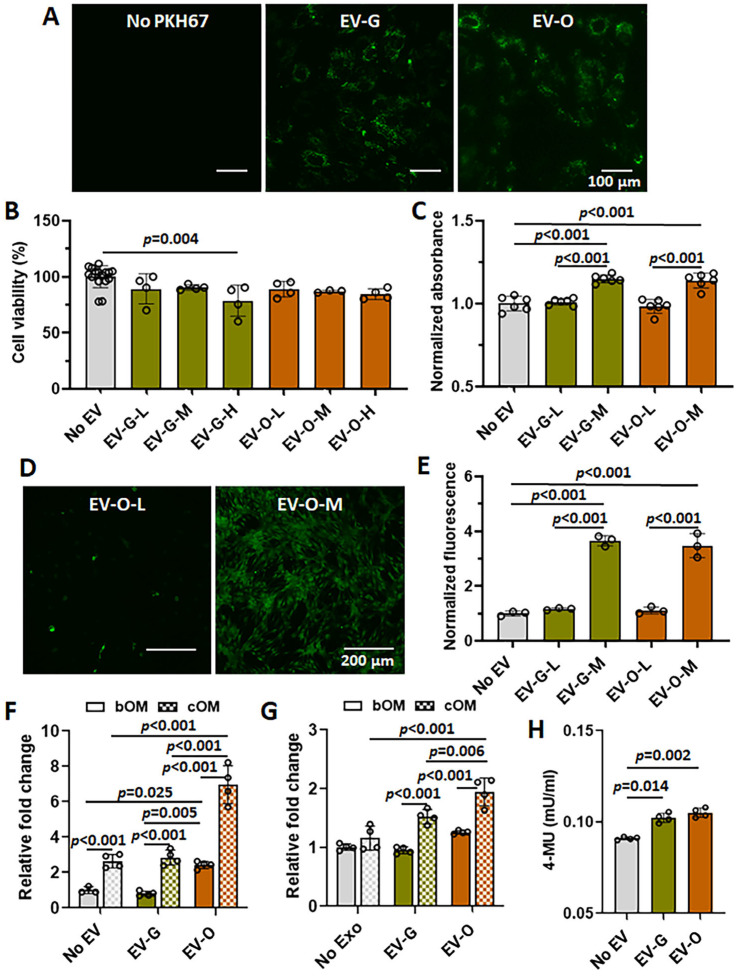
The effect of dental pulp stem cell-derived extracellular vesicles (DPSC-EVs) on cell viability, proliferation, chemotaxis, and odontogenesis. DPSC-EVs were prepared from regular growth medium (EV-G) or odontogenic differentiation medium (EV-O). EV-G-L and EV-O-L: 5 × 10^7^ particles/mL, EV-G-M and EV-O-M: 5 × 10^8^ particles/mL, EV-G-H and EV-O-H: 5 × 10^9^ particles/mL. (**A**) Cellular uptake of DPSCs with or without PKH67 green fluorescence at 2 days. (**B**) Cell viability at 1 day (*n* = 3–16). (**C**) Cell proliferation at 4 days (*n* = 6). (D,E) Chemotaxis at 2 days (*n* = 3): (**D**) representative confocal images of EV-O-M (green: Calcein AM) and (**E**) quantified fluorescence. The confocal images for control and EV-G are available in our previous manuscript [11]. (**F**–**H**) Odontogenic markers at 10 days: (**F**) collagen type 1A1 (COL1A1; *n* = 4), (**G**) runt-related transcription factor 2 (RUNX2; *n* = 4), and (**H**) alkaline phosphatase (ALP; *n* = 4). bOM: basal odontogenic induction medium. cOM: complete odontogenic induction medium. 4-MU: 4-methylumbelliferone. Data are represented as mean ± standard deviation. Colors in bar graphs: gray for No EV, green for EV-G, and orange for EV-O. Lines above bars: *p*-values between two groups.

**Figure 3 ijms-26-07182-f003:**
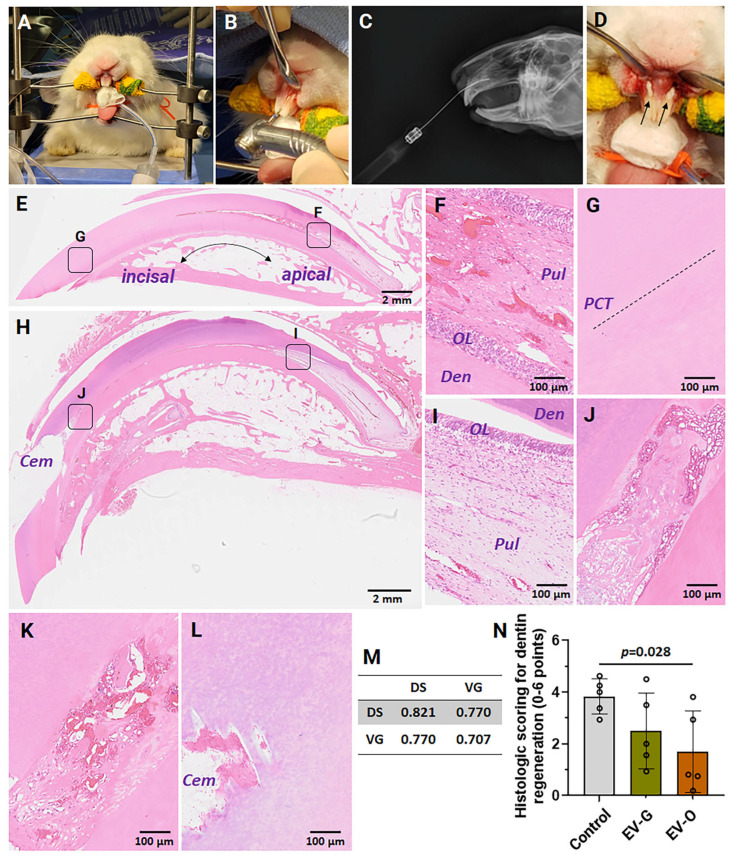
Dentin regeneration in a rabbit incisor partial dentinotomy/pulpotomy model. (**A**) A standard rabbit dental table. (**B**) Cavity creation using a dental bur in the maxillary incisors. (**C**) Validation of pulpal approach using computed tomography (CT). (**D**) Coverage of resin-modified glass ionomer cement after injecting hydrogel only (vehicle control) or extracellular vesicles (EVs)-loaded hydrogel (EV-G or EV-O). (**E**–**L**) Representative histologic images with hematoxylin and eosin stain at 4 weeks. (**E**) Intact mandibular incisor: (**F**) pulp and (**G**) incisal dentin. (**H**) Injured maxillary incisor (vehicle control): (**I**) pulp and (**J**) incisal dentin. (**K**) EV-G. (**L**) EV-O. (**M**) Inter- and intra-observer correlation coefficients. (**N**) Modified histological scoring system for dentin regeneration (*n* = 5). Cem: cement. Den: dentin, OL: odontoblast layer, PCT: pulp cavity trace, Pul: pulp, Control: vehicle control (injury + hydrogel), EV-G: EVs harvested from growth medium, EV-O: EVs harvested from odontogenic differentiation medium. Data are represented as mean ± standard deviation. A line above bars: *p*-value between control and EV-O.

**Table 1 ijms-26-07182-t001:** List of 30 most prevalent mature micro ribonucleic acids (miRNAs). Extracellular vesicles (EVs) were prepared from conditioned media of rabbit (Oryctolagus cuniculus: ocu) dental pulp stem cells under regular growth medium (EV-G) or odontogenic differentiation medium (EV-O).

EV-G			EV-O		
miRNA	Reads	%	miRNA	Reads	%
ocu-miR-146a-5p	296,149.58	13.32	ocu-miR-21-5p	320,322.82	17.51
ocu-miR-199a-3p	217,998.94	9.80	ocu-miR-199a-3p	197,071.34	10.77
ocu-miR-122-5p	190,513.25	8.57	ocu-miR-221-3p	188,002.21	10.28
ocu-miR-221-3p	185,003.32	8.32	ocu-miR-24-3p	90,800.24	4.96
ocu-miR-143-3p	143,493.52	6.45	ocu-miR-214-3p	86,211.23	4.71
ocu-miR-24-3p	137,071.09	6.16	ocu-miR-23b-3p	70,641.32	3.86
ocu-miR-21-5p	122,765.06	5.52	ocu-miR-92a-3p	67,659.80	3.70
ocu-miR-92a-3p	61,986.35	2.79	ocu-miR-29a-3p	65,430.40	3.58
ocu-miR-23b-3p	49,174.99	2.21	ocu-miR-143-3p	63,079.22	3.45
ocu-miR-27b-3p	47,342.73	2.13	ocu-miR-22-3p	58,819.72	3.22
ocu-miR-214-3p	42,716.43	1.92	ocu-miR-27b-3p	47,713.49	2.61
ocu-let-7a-5p	38,945.78	1.75	ocu-miR-125b-5p	38,903.85	2.13
ocu-miR-16b-5p	38,384.95	1.73	ocu-miR-34a-5p	27,086.65	1.48
ocu-miR-378-3p	38,061.67	1.71	ocu-let-7i-5p	24,071.41	1.32
ocu-miR-16a-5p	35,936.55	1.62	ocu-miR-423-5p	21,867.29	1.20
ocu-miR-34a-5p	34,561.72	1.55	ocu-miR-16a-5p	20,585.01	1.13
ocu-miR-93-5p	31,709.98	1.43	ocu-miR-146a-5p	20,118.16	1.10
ocu-miR-152-3p	28,732.49	1.29	ocu-miR-100-5p	19,766.88	1.08
ocu-miR-100-5p	27,396.54	1.23	ocu-miR-152-3p	19,489.91	1.07
ocu-let-7b-5p	25,971.89	1.17	ocu-miR-29c-3p	18,416.89	1.01
ocu-miR-25-3p	25,817.53	1.16	ocu-let-7b-5p	18,332.57	1.00
ocu-miR-29a-3p	23,709.67	1.07	ocu-let-7a-5p	17,266.05	0.94
ocu-miR-423-5p	23,685.38	1.07	ocu-miR-222-3p	17,009.25	0.93
ocu-let-7i-5p	23,552.89	1.06	ocu-miR-423-3p	14,179.10	0.78
ocu-miR-22-3p	21,947.65	0.99	ocu-miR-122-5p	13,182.98	0.72
ocu-let-7f-5p	21,592.76	0.97	ocu-miR-30d-5p	10,777.27	0.59
ocu-miR-6529-5p	17,416.41	0.78	ocu-miR-30e-5p	10,171.98	0.56
ocu-miR-361-5p	15,483.81	0.70	ocu-miR-378-3p	10,125.44	0.55
ocu-miR-128a-3p	14,697.51	0.66	ocu-miR-93-5p	9997.79	0.55
ocu-miR-128b-3p	14,697.51	0.66	ocu-miR-25-3p	8869.77	0.48

**Table 2 ijms-26-07182-t002:** List of significantly up-/down-expressed mature micro ribonucleic acids (miRNAs): EV-O *versus* EV-G in over 0.01% population. Extracellular vesicles (EVs) were prepared from conditioned media of rabbit (Oryctolagus cuniculus: ocu) dental pulp stem cells under regular growth medium (EV-G) or odontogenic differentiation medium (EV-O).

miRNA	log2 (Fold Change)	*p*-Value	miRNA	log2 (Fold Change)	*p*-Value
ocu-miR-708-5p	4.99	1.95 × 10^−13^	ocu-miR-146a-5p	−3.88	0.00084
ocu-miR-708-3p	4.51	4.72 × 10^−14^	ocu-miR-122-5p	−3.85	4.10 × 10^−6^
ocu-miR-885-5p	3.64	2.82 × 10^−13^	ocu-miR-503-5p	−2.45	1.89 × 10^−7^
ocu-miR-24-2-5p	3.40	1.27 × 10^−10^	ocu-miR-16b-5p	−2.18	1.15 × 10^−6^
ocu-miR-29c-3p	3.29	2.13 × 10^−9^	ocu-miR-378-3p	−1.91	0.00022
ocu-miR-874-3p	3.13	7.78 × 10^−10^	ocu-miR-93-5p	−1.67	0.00634
ocu-miR-574-3p	2.80	3.30 × 10^−10^	ocu-miR-127-3p	−1.62	0.00245
ocu-miR-222-3p	2.68	9.00 × 10^−11^	ocu-let-7f-5p	−1.60	0.00120
ocu-miR-125b-5p	2.28	1.63 × 10^−6^	ocu-miR-6529-5p	−1.55	0.00129
ocu-miR-30d-5p	1.91	3.23 × 10^−5^	ocu-miR-25-3p	−1.54	0.00091
ocu-miR-502a-3p	1.76	0.00029	ocu-miR-361-5p	−1.38	0.00283
ocu-miR-181b-5p	1.71	9.27× 10^−5^	ocu-miR-128b-3p	−1.34	0.00193
ocu-miR-342-3p	1.63	0.00018	ocu-miR-128a-3p	−1.34	0.00193
ocu-miR-214-5p	1.62	0.00057	ocu-miR-130b-3p	−1.34	0.00689
ocu-miR-30e-5p	1.57	0.00059	ocu-miR-26b-5p	−1.28	0.00516
ocu-miR-136-3p	1.47	0.00920	ocu-miR-100-3p	−1.26	0.00707
ocu-miR-29a-3p	1.46	0.00226	ocu-miR-143-3p	−1.19	0.02610
ocu-miR-31-5p	1.49	0.00329	ocu-let-7a-5p	−1.17	0.01272
ocu-miR-1296-5p	1.43	0.03054	ocu-miR-151-3p	−1.15	0.01897
ocu-miR-22-3p	1.42	0.00380			
ocu-miR-21-5p	1.38	0.00112			
ocu-miR-365-3p	1.34	0.00225			
ocu-miR-328-3p	1.38	0.01729			
ocu-miR-181a-5p	1.26	0.00350			
ocu-miR-744-5p	1.23	0.01199			
ocu-miR-12090-5p	1.15	0.02734			
ocu-miR-101-3p	1.12	0.00961			
ocu-miR-542-3p	1.12	0.01808			
ocu-miR-29b-3p	1.11	0.01377			
ocu-miR-214-3p	1.01	0.03077			

**Table 3 ijms-26-07182-t003:** A modified histological scoring system for dentin regeneration.

Dentin Formation (0–3)	Inflammatory Cell Response (0–3)
0 = completely covered exposure site	0 = no inflammation at or beneath the exposure site
1 = >50% covered exposure site	1 = few scattered inflammatory cells at or beneath the exposure site
2 = 10–50% covered exposure site	2 = general or localized moderate inflammatory cell infiltration at or beneath the exposure site
3 = no covered exposure site	3 = severe inflammation and/or abscess formation at or beneath the exposure site

## Data Availability

The datasets generated during and/or analyzed during the current study are available from the corresponding author on reasonable request.

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
