# Peer review of "Extracellular Vesicles-Induced Cell Homing and Odontogenesis via microRNA Signaling for Dentin Regeneration"

_ijms, 2025, doi:10.3390/ijms26157182_

Round 1
Reviewer 1 Report
Comments and Suggestions for Authors
The manuscript titled” Extracellular Vesicles-Based Cell Homing and Odontogenic Differentiation for Dentin Regeneration and Their Profiles of microRNAs” presents a well-designed and comprehensive preclinical study evaluating the effects of extracellular vesicles (EVs) derived from dental pulp stem cells (DPSCs) on cell homing and odontogenic differentiation. The authors successfully demonstrate that EVs, especially those derived under odontogenic induction conditions (EV-O), enhance DPSC chemotaxis, proliferation, and differentiation both in vitro and in vivo. Furthermore, the study provides insightful miRNA profiling of EVs and identifies key candidates potentially responsible for these regenerative effects.
The study is technically sound and methodologically rigorous, incorporating both functional and molecular analyses.
Abstract - well written and informative.
Introduction - it clearly defines the rationale for this study, although it would benefit from more explanation of the knowledge gap and more focused justification for miRNA profiling.
Materials and methods section is detailed and technically appropriate. NGS analysis methods and EV dosing schemes should be clarified further for reproducibility. Clarify all EV concentrations early in the methods and standardize abbreviations (EV-L, EV-M, etc.).
Results section - although EV size and concentration are reported, additional metrics (e.g., particle/protein ratio, yield per cell) would improve reproducibility and comparability to other EV studies.
Discussion section-
While the miRNA data are comprehensive, the manuscript lacks sufficient explanation of how the most significantly altered miRNAs (e.g., miR-708, miR-146a, miR-29b) may functionally relate to dentinogenesis, inflammation, or stem cell recruitment.
The limitations of the rabbit incisor model (e.g., continuous growth, open root) should be more explicitly addressed in the Discussion, along with recommendations for alternative models or earlier time points.
In addtion, the role of individual miRNAs is speculative; although acknowledged in the limitations, this weakens mechanistic conclusions. Future validation via qRT-PCR or knockdown experiments should be suggested more explicitly.
Also, the authors should revise overly confident statements ("confirmed", "demonstrated") to reflect preclinical nature. Replace them with ”sugested”, ”provides evidence”, etc.
Conclusion section should acknowledge limitations more explicitly and temper confident language.
Author Response
Reviewer(s) Comments and Suggestions for Authors:
Reviewer #1
The manuscript titled” Extracellular Vesicles-Based Cell Homing and Odontogenic Differentiation for Dentin Regeneration and Their Profiles of microRNAs” presents a well-designed and comprehensive preclinical study evaluating the effects of extracellular vesicles (EVs) derived from dental pulp stem cells (DPSCs) on cell homing and odontogenic differentiation. The authors successfully demonstrate that EVs, especially those derived under odontogenic induction conditions (EV-O), enhance DPSC chemotaxis, proliferation, and differentiation both in vitro and in vivo. Furthermore, the study provides insightful miRNA profiling of EVs and identifies key candidates potentially responsible for these regenerative effects.
Comments:
The study is technically sound and methodologically rigorous, incorporating both functional and molecular analyses.
Abstract - well written and informative.
1. Introduction - it clearly defines the rationale for this study, although it would benefit from more explanation of the knowledge gap and more focused justification for miRNA profiling.
The introduction was modified according to your comment.
(Old) Extracellular vesicles (EVs) including exosomes (30 – 150 nm size range), microvesicles (MVs; 100 – 1,000 nm size range), and apoptotic bodies have attracted attention in regenerative medicine for their remarkable potential to promote cell recruitment, proliferation, and differentiation through intercellular communication [5-9]. In our previous studies, EVs derived from DPSCs (DPSC-EVs) under angiogenic differentiation conditions (EV-A) enhanced DSPC chemotaxis, proliferation, and angiogenic differentiation [10]. Similarly, EVs isolated from odontogenically differentiated DPSCs (EV-O) elevated the expression of odontogenic genes in vitro through transforming growth factor beta 1 (TGFβ1)/mothers against decapentaplegic homolog (Smad) pathway [11]. However, the effects of EV-O on cell homing were not explored [11].
Among a variety of bioactive molecules in EVs such as deoxyribonucleic acid (DNA), ribonucleic acid (RNA), lipids, and proteins [12], microRNAs (miRNAs/miRs) play a crucial role in regulating gene expression and cellular processes [13]. For example, miR-27a-5p mimics over-expressed in human EV-O up-regulated odontogenic markers, dentin sialophosphoprotein (DSPP), dentin matrix protein 1 (DMP-1), alkaline phosphatase (ALP), and runt-related transcription factor 2 (RUNX2) [11]. Thus, miRNA sequencing can allow understanding the functional mechanisms of EVs and targeting potential miRNAs.
(New: lines 47-67) Extracellular vesicles (EVs) including exosomes (30 – 150 nm size range), microvesicles (MVs; 100 – 1,000 nm size range), and apoptotic bodies have attracted attention in regenerative medicine for their remarkable potential to promote cell recruitment, proliferation, and differentiation through intercellular communication [5-9]. In the previous study, EVs isolated from odontogenically differentiated human DPSCs (EV-O) elevated the expression of odontogenic genes in vitro through transforming growth factor beta 1 (TGFβ1)/mothers against decapentaplegic homolog (Smad) pathway [10]. However, the effects of EV-O on cell homing were not explored [10]. Our previous studies revealed that EVs derived from DPSCs under growth (EV-G) and angiogenic differentiation conditions (EV-A) promoted approximately a 4-fold increase over control in DPSC migration [11]. Thus, EV-O can be a great source to enhance both DPSC homing and odontogenesis for pulpodentin regeneration.
Among a variety of bioactive molecules in EVs such as deoxyribonucleic acid (DNA), ribonucleic acid (RNA), lipids, and proteins [12], microRNAs (miRNAs/miRs) play a crucial role in regulating gene expression and cellular processes [13]. During tooth development or repair, several miRNAs are strongly associated with odontogenic differentiation for dentin formation [10,14,15]. For example, miR-27a-5p mimics over-expressed in human EV-O up-regulated odontogenic markers, dentin sialophosphoprotein (DSPP), dentin matrix protein 1 (DMP-1), alkaline phosphatase (ALP), and runt-related transcription factor 2 (RUNX2) [10]. Thus, miRNA sequencing can allow understanding of the functional mechanisms of EVs and targeting potential miRNAs.
2. Materials and methods section is detailed and technically appropriate. NGS analysis methods and EV dosing schemes should be clarified further for reproducibility. Clarify all EV concentrations early in the methods and standardize abbreviations (EV-L, EV-M, etc.).
2.1. NGS analysis is updated in terms of sample numbers and data analysis.
(Old) RNAs were extracted from two batches of rabbit DPSC-EVs (EV-G and EV-O) and quantified by a Bioanalyzer Small RNA Assay kit (Agilent, Santa Clara, CA, USA). NGS libraries were prepared and sequenced on a HiSeq® Sequencing System (Illumina, San Diego, CA, USA) with 150 bp paired-end reads at an approximate depth of 10-15 million reads per sample in System Biosciences [10,42]. Raw data were analyzed in the Bioinformatics Division of the Iowa Institute of Human Genetics (IIHG) according to the previous protocol [10].
(New: Lines 322-331) RNAs were extracted from biological duplicates of rabbit DPSC-EVs (EV-G and EV-O) and quantified by a Bioanalyzer Small RNA Assay kit (Agilent, Santa Clara, CA, USA). NGS libraries were prepared and sequenced on a HiSeq® Sequencing System (Illumina, San Diego, CA, USA) with 150 bp paired-end reads at an approximate depth of 10-15 million reads per sample in System Biosciences [11,44]. Raw data were analyzed in the Bioinformatics Division of the Iowa Institute of Human Genetics (IIHG) according to the previous protocol [11]. In brief, the workflow used Trim Galore! (version 0.6.6) for adapter trimming and Bowtie1 (version 1.3.0) for aligning reads to small RNA sequences. Read counts were imported into R (version 3.5.2). Count normalization and statistical analysis were performed using DESeq2 (version 1.22.2).
2.2. Clarified the abbreviations in the methods and the figure 2 legend. For example, EV-L => EV-G-L and EV-O-L.
3. Results section - although EV size and concentration are reported, additional metrics (e.g., particle/protein ratio, yield per cell) would improve reproducibility and comparability to other EV studies.
We agree that further characterization beyond size and concentration is warranted for the reasons cited and plan to include such metrics in future studies. This issue is mentioned as a limitation in the discussion (lines 261-262).
4. Discussion section
4.1. While the miRNA data are comprehensive, the manuscript lacks sufficient explanation of how the most significantly altered miRNAs (e.g., miR-708, miR-146a, miR-29b) may functionally relate to dentinogenesis, inflammation, or stem cell recruitment.
The potential roles related to odontogenesis and anti-inflammation were updated with additional references based on the previous reports.
(Old) Instead, ocu-miR-708-3p (4.51-fold higher log2 fold change in EV-O: 4.51Δ) [33], ocu-miR-22-3p (1.42Δ) [34], ocu-miR-21-5p (1.38Δ) [35], and ocu-miR-29b-3p (1.11Δ) [36,37] were significantly up-regulated in EV-O (Table 2) and known to support odontogenesis or osteogenesis in the previous studies.
(New: Lines 240-244) Instead, our novel findings were significant up-regulation of ocu-miR-708-3p (4.51-fold higher log2 fold change in EV-O: 4.51Δ) [35,36], ocu-miR-22-3p (1.42Δ) [37], ocu-miR-21-5p (1.38Δ) [38], and ocu-miR-29b-3p (1.11Δ) [39-41] in EV-O (Table 2), and these miRNAs are known to promote odontogenesis/osteogenesis and to exert anti-inflammation in the previous studies.
4.2. The limitations of the rabbit incisor model (e.g., continuous growth, open root) should be more explicitly addressed in the Discussion, along with recommendations for alternative models or earlier time points.
The statement was edited accordingly.
(Old) One of the possible reasons may be the continuous growth of rabbit incisors (approximately 2 mm per week) with open roots, where numerous stem cells are present [39]. Therefore, future studies should assess earlier time-points (1 or 2 weeks).
(New: Lines 251-254) One of the possible reasons may be the continuous growth of rabbit incisors (approximately 2 mm per week) with open roots, where numerous stem cells are present [43]. This factor may lead to an overestimation of regeneration potential compared to human teeth. Therefore, future studies should assess earlier time-points (1 or 2 weeks).
4.3. In addition, the role of individual miRNAs is speculative; although acknowledged in the limitations, this weakens mechanistic conclusions. Future validation via qRT-PCR or knockdown experiments should be suggested more explicitly.
I understand your concern. Currently, we are preparing a grant application for further studies of target miRNAs. This manuscript will be a great reference as preliminary data.
4.4. Also, the authors should revise overly confident statements ("confirmed", "demonstrated") to reflect preclinical nature. Replace them with ”suggested”, ”provides evidence”, etc.
As suggested, some words of "confirmed" and "demonstrated" were replaced with ” suggested” and ”provides evidence” in the Discussion.
5. Conclusion section should acknowledge limitations more explicitly and temper confident language.
The statement was edited accordingly.
(Old) With the limitations of this study, these results suggest that our EV-based strategy holds strong therapeutic potential for dentin regeneration by promoting both cell recruitment and odontogenesis.
(New: Lines 389-391) Thus, our EV-based strategy holds therapeutic potential for dentin regeneration by promoting both cell recruitment and odontogenesis.

Reviewer 2 Report
Comments and Suggestions for Authors
With interest I’ve read the paper “Extracellular Vesicles-Based Cell Homing and Odontogenic Differentiation for Dentin Regeneration and Their Profiles of microRNAs.” In this in vitro and in vivo study, the authors assessed the effects of extracellular vesicles on pulp stem cells homing and differentiation for dentin regeneration in rabbits.
The article is novel, interesting, and sound; however, one minor comment could be addressed.
In the introduction, more insights should be given regarding precious studies on the effects of DPSC on pulp regeneration (especially odontogenic differentiation) to provide a wider background of what is already done in the field. Also, the novelty of the study should be highlighted better.
Author Response
Reviewer(s) Comments and Suggestions for Authors:
Reviewer #2
With interest I’ve read the paper “Extracellular Vesicles-Based Cell Homing and Odontogenic Differentiation for Dentin Regeneration and Their Profiles of microRNAs.” In this in vitro and in vivo study, the authors assessed the effects of extracellular vesicles on pulp stem cells homing and differentiation for dentin regeneration in rabbits. The article is novel, interesting, and sound; however, one minor comment could be addressed.
Comments:
In the introduction, more insights should be given regarding previous studies on the effects of DPSC on pulp regeneration (especially odontogenic differentiation) to provide a wider background of what is already done in the field. Also, the novelty of the study should be highlighted better.
The introduction was modified and rearranged with additional statements to emphasize study rationale and novelty.
(Old) In our previous studies, EVs derived from DPSCs (DPSC-EVs) under angiogenic differentiation conditions (EV-A) enhanced DSPC chemotaxis, proliferation, and angiogenic differentiation [10]. Similarly, EVs isolated from odontogenically differentiated DPSCs (EV-O) elevated the expression of odontogenic genes in vitro through transforming growth factor beta 1 (TGFβ1)/mothers against decapentaplegic homolog (Smad) pathway [11]. However, the effects of EV-O on cell homing were not explored [11].
(New: lines 50-58) In the previous study, EVs isolated from odontogenically differentiated human DPSCs (EV-O) elevated the expression of odontogenic genes in vitro through transforming growth factor beta 1 (TGFβ1)/mothers against decapentaplegic homolog (Smad) pathway [10]. However, the effects of EV-O on cell homing were not explored [10]. Our previous studies revealed that EVs derived from DPSCs under growth (EV-G) and angiogenic differentiation conditions (EV-A) promoted approximately a 4-fold increase over control in DPSC migration [11]. Thus, EV-O can be a great source to enhance both DPSC homing and odontogenesis for pulpodentin regeneration.

Reviewer 3 Report
Comments and Suggestions for Authors
The manuscript titled “Extracellular Vesicles-Based Cell Homing and Odontogenic Differentiation for Dentin Regeneration and Their Profiles of microRNAs”,
The manuscript addresses a timely and significant topic in regenerative dentistry using DPSC-derived EVs. The dual focus on cell homing and odontogenic differentiation is well-justified. The inclusion of both in vitro and in vivo studies adds strength.
Title and Abstract
- Lines 1–3: The title is clear but long. Consider:
Suggested Title: “Extracellular Vesicle-Induced Cell Homing and Odontogenesis via miRNA Signaling for Dentin Regeneration” - Lines 21–36 (Abstract):
Line 29: Add control comparison for "5x10⁸ EVs/mL" to clarify significance.
Line 30–32: Rephrase “resulted in approximately 55% greater dentin regeneration...” for clarity. Suggest:
"EV-O enhanced dentin regeneration by approximately 55% over controls in a rabbit partial dentinotomy model." - Line 33: Specify which miRNAs (e.g., miR-21-5p, miR-708) were most relevant.
Introduction
- Line 41–45: Sentence beginning “Odontoblasts…” is wordy. Suggested:
“Odontoblasts, located in the dental pulp, secrete and maintain dentin. When damaged, they are replaced by odontoblast-like cells derived from DPSCs.” - Lines 56–57: Add a citation after “were not explored.”
- Line 66: "evaluate the effects of DPSC-EVs" → clarify you compare EV-G and EV-O.
Results
- Line 74–76: Clarify if the difference in EV size was statistically significant.
- Line 100–104:
Typo: “EV-H concentration was excluded…” — consider explaining why only the highest dose reduced viability.
Line 104: Suggest rephrasing:
“Due to cytotoxicity observed at the highest concentration (EV-H), only EV-M and EV-L were assessed further.” - Line 108: “promoted dramatic cell migration” → suggest more precise wording:
“induced a significant increase in DPSC migration” - Lines 112–118: Clarify if effects are additive with cOM + EV-O, or if EV-O alone is sufficient.
miRNA Profiles
- Lines 134–152:
Suggest highlighting key differences between EV-G and EV-O in miRNA expression in narrative form before diving into Table 1.
Line 138: “more than 8% of the population…” — population of what? Reads? Clarify wording.
In Vivo
- Lines 157–172:
Line 164–165: "Fatty-like soft tissues..."—consider revising to more scientific terminology.
Line 169: Correlation coefficient values (>0.7) should include actual values or statistical detail (e.g., “τ = 0.81, p < 0.01”).
Discussion
- Line 187: Add a stronger opening statement for discussion. Suggest:
“Our findings demonstrate that DPSC-derived EVs, particularly EV-O, have the potential to activate key processes in dentin repair by promoting cell migration and differentiation.” - Lines 231–241: Provide a clearer contrast between miRNAs identified in this study and prior literature.
Line 238: "Instead, ocu-miR-708-3p..." → emphasize novelty. - Lines 249–252: The discussion about rabbit incisor growth rate is important. However, suggest emphasizing how this might limit the model rather than just noting it.
Suggest adding: “These factors may overestimate regeneration potential compared to human teeth.”
Conclusions
- Lines 371–379:
Well summarized. Still, you may consider shortening slightly and listing the top 2–3 candidate miRNAs clearly for clinical translation.
Methods
- Lines 265–285:
Clarify any quality control (e.g., EV purity, absence of protein contaminants).
Line 281: Describe any validation of NTA results (e.g., replicate runs). - Lines 316–323:
Line 319: Add read mapping or filtering criteria.
Add a note on whether miRNA expression was normalized and which statistical tools were used.
Graphic Abstract: Consider adding a clear schematic showing EV collection, uptake by DPSCs, signaling, and dentin regeneration in vivo.
Supplementary Figures: Include a volcano plot or heatmap of miRNA differential expression (Table 2 is dense).
Statistical Power: Address in the discussion if the number of replicates (especially n = 5 in vivo) is adequate.
Reproducibility: Mention if EV isolation was done in technical triplicates or biological duplicates for miRNA.
Author Response
Reviewer(s) Comments and Suggestions for Authors:
Reviewer #3
The manuscript titled “Extracellular Vesicles-Based Cell Homing and Odontogenic Differentiation for Dentin Regeneration and Their Profiles of microRNAs”,
The manuscript addresses a timely and significant topic in regenerative dentistry using DPSC-derived EVs. The dual focus on cell homing and odontogenic differentiation is well-justified. The inclusion of both in vitro and in vivo studies adds strength.
Comments:
1. (Title) Lines 1–3: The title is clear but long. Consider:
Suggested Title: “Extracellular Vesicle-Induced Cell Homing and Odontogenesis via miRNA Signaling for Dentin Regeneration”
The title was edited according to your suggestion with minor changes.
(Old) Extracellular Vesicles-Based Cell Homing and Odontogenic Differentiation for Dentin Regeneration and Their Profiles of microRNAs
(New) Extracellular Vesicles-Induced Cell Homing and Odontogenesis via microRNA Signaling for Dentin Regeneration
2. (Abstract) Line 29: Add control comparison for "5x10⁸ EVs/mL" to clarify significance.
The information about the comparison groups was added.
(Old) Treatment with 5x108 EVs/mL significantly enhanced DPSC chemotaxis and proliferation.
(New: lines 28-29) Treatment with 5x108 EVs/mL significantly enhanced DPSC chemotaxis and proliferation compared with no-treatment control and a lower dosage of EV (5x107 EVs/mL).
3. (Abstract) Line 30–32: Rephrase “resulted in approximately 55% greater dentin regeneration...” for clarity. Suggest: "EV-O enhanced dentin regeneration by approximately 55% over controls in a rabbit partial dentinotomy model."
The statement is edited according to your suggestion with minor changes.
(Old) In a partial dentinotomy/pulpotomy model, EV-O treatment resulted in approximately 55% greater dentin regeneration compared to vehicle controls.
(New: lines 32-33) EV-O enhanced dentin regeneration by approximately 55% over vehicle controls in a rabbit partial dentinotomy/pulpotomy model.
4. (Abstract) Line 33: Specify which miRNAs (e.g., miR-21-5p, miR-708) were most relevant.
Specific lists of miRNAs were added.
(Old) We identified key microRNAs in EV-O involved in cell homing and odontogenesis.
(New: lines 33-34) We identified key microRNAs (miR-21-5p, miR-221-3p, and miR-708-3p) in EV-O involved in cell homing and odontogenesis.
5. (Introduction) Line 41–45: Sentence beginning “Odontoblasts…” is wordy. Suggested:
“Odontoblasts, located in the dental pulp, secrete and maintain dentin. When damaged, they are replaced by odontoblast-like cells derived from DPSCs.”
The statement is edited according to your suggestion with minor changes.
(Old) Odontoblasts, located in the outer layer of the dental pulp, are specialized cells responsible for dentin formation and maintenance. When the dentin is severely damaged by caries, wear, or fractures, odontoblasts may be destroyed and replaced by odontoblast-like cells derived from dental pulp stem cells (DPSCs).
(New: lines 42-44) Odontoblasts, located in the outer layer of the dental pulp, are responsible for dentin formation and maintenance. When damaged, they are replaced by odontoblast-like cells derived from dental pulp stem cells (DPSCs).
6. (Introduction) Lines 56–57: Add a citation after “were not explored.”
A citation was added.
7. (Introduction) Line 66: "evaluate the effects of DPSC-EVs" → clarify you compare EV-G and EV-O.
The statement is edited according to your suggestion.
(Old) Here, we hypothesized that EVs isolated from odontogenic-differentiated DPSCs have unique miRNA profiles to enhance DPSC chemotaxis and odontogenesis. The objective of this study was to evaluate the effects of DPSC-EVs on cell homing and odontogenesis for dentin regeneration via both in vitro and in vivo studies.
(New: lines 68-71) We hypothesized that EV-O has unique miRNA profiles to enhance DPSC chemotaxis and odontogenesis. The objectives of this study were to evaluate the effects of DPSC-EVs and to compare EV-G and EV-O on cell homing and odontogenesis for dentin regeneration via both in vitro and in vivo studies.
8. (Results) Line 74–76: Clarify if the difference in EV size was statistically significant.
A p-value (p < 0.001) was added (line 78).
9. (Results) Line 100–104: Typo: “EV-H concentration was excluded…” — consider explaining why only the highest dose reduced viability. Suggest rephrasing: “Due to cytotoxicity observed at the highest concentration (EV-H), only EV-M and EV-L were assessed further.”
The statement is edited according to your suggestion with minor changes.
(Old) The EV-H concentration was excluded for further evaluation.
(New: lines 104-105) Due to cytotoxicity observed at the highest concentrations, EV-G-H and EV-O-H were excluded for further evaluation.
10. (Results) Line 108: “promoted dramatic cell migration” → suggest more precise wording:
“induced a significant increase in DPSC migration”
The statement is edited according to your suggestion with minor changes.
(Old) In a chemotactic assay, DPSC-EV treatment with a concentration of 5 x 108 particle/mL promoted dramatic cell migration in both groups of EV-G (p < 0.001 versus No EV or EV-G-L) and EV-O (p < 0.001 versus No EV or EV-O-L) (Figure 2D and 2E).
(New: lines 108-111) In a chemotactic assay, DPSC-EV treatment with a concentration of 5 x 108 particles/mL promoted a significant increase of DPSC migration in both groups of EV-G (p < 0.001 versus No EV or EV-G-L) and EV-O (p < 0.001 versus No EV or EV-O-L) (Figure 2D and 2E).
11. (Results) Lines 112–118: Clarify if effects are additive with cOM + EV-O, or if EV-O alone is sufficient.
A statement is added to clarify the additive effect of cOM.
(Old) Similarly, EV-O induced higher expression of RUNX2 (p < 0.001 versus No EV and p = 0.006 versus EV-G) (Figure 2G).
(New: lines 117-119) Similarly, EV-O induced higher expression of RUNX2 in only cOM (p < 0.001 versus No EV and p = 0.006 versus EV-G) (Figure 2G). Thus, the odontogenic effects of EV-O were additive in the cOM condition.
12. (Results) Lines 134–152: Suggest highlighting key differences between EV-G and EV-O in miRNA expression in narrative form before diving into Table 1.
A statement is added to emphasize the purpose of miRNA profiles.
(Old) In NGS analysis, we found a total of 474 mature and 254 hairpin miRNAs in EV-G and a total of 431 mature and 256 hairpin miRNAs in EV-O.
(New: lines 136-138) In NGS analysis, we identified candidate miRNAs that regulate cell homing and odontogenesis in DPSC-EVs. A total of 474 mature and 254 hairpin miRNAs in EV-G and 431 mature and 256 hairpin miRNAs in EV-O were detected.
13. (Results) Line 138: “more than 8% of the population…” — population of what? Reads? Clarify wording.
8% of the population => 8% of the total reads (line 142)
14. (Results) Lines 164–165: "Fatty-like soft tissues..."—consider revising to more scientific terminology.
Although we observed adipose-like tissue formation, the statement was removed to emphasize inflammatory responses.
(Old) Fatty-like soft tissues with inflammatory responses were observed through the pulp cavity trace (PCT) (Figure 3J).
(New: line 168) Inflammatory responses were observed through the pulp cavity trace (PCT) (Figure 3J).
15. (Results) Line 169: Correlation coefficient values (>0.7) should include actual values or statistical detail (e.g., “τ = 0.81, p < 0.01”).
Actual values are available in Figure 3M.
16. (Discussion) Line 187: Add a stronger opening statement for discussion. Suggest:
“Our findings demonstrate that DPSC-derived EVs, particularly EV-O, have the potential to activate key processes in dentin repair by promoting cell migration and differentiation.”
The suggested statement was added.
(Old) The goal of this study was to evaluate the therapeutic potential of DPSC-EVs to enhance cell homing and odontogenic differentiation for dentin regeneration. EVs were isolated from DPSCs cultured under growth (EV-G) or odontogenic differentiation (EV-O) conditions, and their miRNA profiles were analyzed.
(New: lines 190-191) Our findings provide evidence that DPSC-EVs, particularly EV-O, have the potential to activate key processes in dentin repair by promoting cell homing and differentiation.
17. (Discussion) Lines 231–241: Provide a clearer contrast between miRNAs identified in this study and prior literature.
A major difference in miR-27a-5p expression was compared: 11-fold up in Hu et al versus >0.01% total reads in this study.
(Old) However, our miRNA profiles differed from Hu and colleagues [11]. The expression of miR-27a-5p, which promoted odontogenic differentiation of DPSCs through transforming growth factor beta 1 (TGFβ1), was less than 0.01% population in our dataset, compared to 11-fold higher in their study [11].
(New: lines 236-240) However, our miRNA profiles differed from Hu and colleagues who used human DPSCs as an EV source [10]. The expression of miR-27a-5p, which promoted odontogenic differentiation of DPSCs through transforming growth factor beta 1 (TGFβ1), was less than 0.01% reads in our dataset, compared to 11-fold higher in their study [10].
18. (Discussion) Line 238: "Instead, ocu-miR-708-3p..." → emphasize novelty.
A statement is edited to emphasize our novel findings.
(Old) Instead, ocu-miR-708-3p (4.51-fold higher log2 fold change in EV-O: 4.51Δ) [33], ocu-miR-22-3p (1.42Δ) [34], ocu-miR-21-5p (1.38Δ) [35], and ocu-miR-29b-3p (1.11Δ) [36,37] were significantly up-regulated in EV-O (Table 2) and known to support odontogenesis or osteogenesis in the previous studies.
(New: lines 240-244) Instead, our novel findings were significant up-regulation of ocu-miR-708-3p (4.51-fold higher log2 fold change in EV-O: 4.51Δ) [35,36], ocu-miR-22-3p (1.42Δ) [37], ocu-miR-21-5p (1.38Δ) [38], and ocu-miR-29b-3p (1.11Δ) [39-41] in EV-O (Table 2), and these miRNAs are known to promote odontogenesis/osteogenesis and to exert anti-inflammation in the previous studies.
19. (Discussion) Lines 249–252: The discussion about rabbit incisor growth rate is important. However, suggest emphasizing how this might limit the model rather than just noting it.
Suggest adding: “These factors may overestimate regeneration potential compared to human teeth.”
As suggested, a statement was added.
(New: lines 253-254) This factor may lead to an overestimation of regeneration potential compared to human teeth.
20. (Conclusions) Lines 371–379: Well summarized. Still, you may consider shortening slightly and listing the top 2–3 candidate miRNAs clearly for clinical translation.
The section of conclusions was edited accordingly.
(Old) In addition, we identified candidate miRNAs that regulate cell homing (miR-21-5p, miR-221-3p, miR-214-3p, miR-29a-3p, and miR-152-3p) and odontogenesis (miR-708-3p, miR-22-3p, miR-21-5p, and miR-29b-3p). With the limitations of this study, these results suggest that our EV-based strategy holds strong therapeutic potential for dentin regeneration by promoting both cell recruitment and odontogenesis.
(New: lines 388-391) In addition, we identified candidate miRNAs that regulate cell homing (miR-21-5p and miR-221-3p) and odontogenesis (miR-708-3p). Thus, our EV-based strategy holds therapeutic potential for dentin regeneration by promoting both cell recruitment and odontogenesis.
21. (Methods) Lines 265–285: Clarify any quality control (e.g., EV purity, absence of protein contaminants).
Quality control was evaluated by the characterization of DPSC-EVs shown in Figure 1. In particular, the absence of protein contamination was confirmed in the antibody array (Figure 1D), such as GM130.
22. (Methods) Line 281: Describe any validation of NTA results (e.g., replicate runs).
The information of replicate runs (n = 3) is in the Figure 1A legend.
23. (Methods) Line 319: Add read mapping or filtering criteria. Add a note on whether miRNA expression was normalized and which statistical tools were used.
The following information was added.
(New: lines 328-331) In brief, the workflow used Trim Galore! (version 0.6.6) for adapter trimming and Bowtie1 (version 1.3.0) for aligning reads to small RNA sequences. Read counts were imported into R (version 3.5.2). Count normalization and statistical analysis were performed using DESeq2 (version 1.22.2).
24. Graphic Abstract: Consider adding a clear schematic showing EV collection, uptake by DPSCs, signaling, and dentin regeneration in vivo.
Since the information was introduced in the previous manuscript (see a graphic abstract below for our previous manuscript: https://www.mdpi.com/1422-0067/24/1/466), we want to mainly emphasize the strategy of cell homing and differentiation in this manuscript.

25. Supplementary Figures: Include a volcano plot or heatmap of miRNA differential expression (Table 2 is dense).
Volcano plots for mature and hairpin miRNAs were added as a supplementary figure. Table 2 was extended.

Figure S1.
26. Statistical Power: Address in the discussion if the number of replicates (especially n = 5 in vivo) is adequate.
Unlike in vitro data, the statistical power of in vivo data was less than 0.8. Therefore, a statement of “insufficient replicates for power analysis in the animal studies” was added in the discussion (lines 264-265).
27. Reproducibility: Mention if EV isolation was done in technical triplicates or biological duplicates for miRNA.
two batches => biological duplicates (line 322)

Round 2
Reviewer 1 Report
Comments and Suggestions for Authors
The manuscript is now improved and can be considered for publication
Reviewer 3 Report
Comments and Suggestions for Authors
Manuscript ready for publication